# Early moderate prenatal alcohol exposure and maternal diet impact offspring DNA methylation across species

**Mitchell Bestry[1], Alexander N Larcombe[2,3], Nina Kresoje[4], Emily K Chivers[2], Chloe Bakker[2], James P Fitzpatrick[5], Elizabeth J Elliott[6,7], Jeffrey M Craig[8,9,10], Evelyne Muggli[9,10], Jane Halliday[10,11], Delyse Hutchinson[11,12,13,14], Sam Buckberry[4,15,16], Ryan Lister[15,16], Martyn Symons[1,17], David Martino[18]***

[1]Telethon Kids Institute, The University of Western Australia, Perth, Australia; [2]Respiratory Environmental Health, Wal-yan Respiratory Research Centre, Telethon Kids Institute, Nedlands, Australia; [3]Occupation, Environment and Safety, School of Population Health, Curtin University, Perth, Australia; [4]Telethon Kids Institute, Nedlands, Australia; [5]School of Psychological Sciences, University of Western Australia, Perth, Australia; [6]University of Sydney, Faculty of Medicine and Health, Specialty of Child and Adolescent Health, Melbourne, Australia; [7]Sydney Children's Hospitals Network (Westmead) and Kids Research, Geelong, Australia; [8]Deakin University, IMPACT – the Institute for Mental and Physical Health and Clinical Translation, School of Medicine, Geelong, Australia; [9]Murdoch Children's Research Institute, Royal Children's Hospital, Parkville, Australia; [10]Department of Paediatrics, University of Melbourne, Royal Children's Hospital, Parkville, Australia; [11]Reproductive Epidemiology, Murdoch Children's Research Institute, Parkville, Australia; [12]Deakin University, School of Psychology, Faculty of Health, Geelong, Australia; [13]Murdoch Children's Research Institute, Centre for Adolescent Health, Royal Children's Hospital, Melbourne, Australia; [14]University New South Wales, National Drug and Alcohol Research Centre, Sydney, Australia; [15]Harry Perkins Institute of Medical Research, QEII Medical Centre and Centre for Medical Research, The University of Western Australia, Perth, Australia; [16]ARC Centre of Excellence in Plant Energy Biology, School of Molecular Sciences, The University of Western Australia, Perth, Australia; [17]National Drug Research Institute, enAble Institute, Curtin University, Perth, Australia; [18]Wal-yan Respiratory Research Centre, Telethon Kids Institute, Nedlands, Australia

*For correspondence:
david.martino@telethonkids.
org.au

**Competing interest:** The authors declare that no competing interests exist.

**Abstract** Alcohol consumption in pregnancy can affect genome regulation in the developing offspring but results have been contradictory. We employed a physiologically relevant murine model of short-term moderate prenatal alcohol exposure (PAE) resembling common patterns of alcohol consumption in pregnancy in humans. Early moderate PAE was sufficient to affect site-specific DNA methylation in newborn pups without altering behavioural outcomes in adult littermates. Whole-genome bisulfite sequencing of neonatal brain and liver revealed stochastic influence on DNA methylation that was mostly tissue-specific, with some perturbations likely originating as early as gastrulation. DNA methylation differences were enriched in non-coding genomic regions with regulatory potential indicative of broad effects of alcohol on genome regulation. Replication studies in human cohorts with fetal alcohol spectrum disorder suggested some effects were metastable at genes linked to disease-relevant traits including facial morphology, intelligence, educational

attainment, autism, and schizophrenia. In our murine model, a maternal diet high in folate and choline protected against some of the damaging effects of early moderate PAE on DNA methylation. Our studies demonstrate that early moderate exposure is sufficient to affect fetal genome regulation even in the absence of overt phenotypic changes and highlight a role for preventative maternal dietary interventions.

## eLife assessment

This **important** study unveils the significant impact of prenatal alcohol exposure on epigenetic patterns, offering new insights into its adverse health outcomes through **solid** evidence from both mouse models and human data. The findings, which reveal how a high-methyl diet can mitigate these epigenetic alterations, present a promising prenatal care strategy. Despite its **solid** data overall, the study's small sample size and unaccounted confounders suggest the need for further research to confirm these findings and explore their practical implications.

## Introduction

Alcohol consumption in pregnancy is a leading cause of neurodevelopmental impairments in children (*Popova et al., 2023*). Alcohol can pass through the placenta acting as a teratogen in fetal tissues causing physical, cognitive, behavioural, and neurodevelopmental impairment in children at high doses with lifelong consequences for health (*Chung et al., 2021*). Fetal alcohol spectrum disorder (FASD) and fetal alcohol syndrome (FAS) can arise at binge levels of exposure, although not always at lower levels of exposure. Whether prenatal alcohol exposure (PAE) is sufficient to induce overt physiological abnormalities depends on multiple environmental and genetic factors including the dose and timing of alcohol use during pregnancy, maternal diet, smoking, stress, and potentially other factors that collectively influence fetal outcomes (*Chung et al., 2021*; *Jacobson et al., 2021*; *Sambo and Goldman, 2023*).

Patterns of alcohol consumption in pregnancy vary, but epidemiological surveys suggest most women in Western countries consume low to moderate levels between conception until recognition of pregnancy (*Tsang et al., 2022*), after which time consumption largely ceases, apart from occasional use (*McCormack et al., 2017*). While the effects of binge levels of exposure are well documented as able to cause FASD, more subtle effects that reflect the more common patterns of drinking are unclear and more research is needed to support public health initiatives to reduce alcohol consumption in pregnancy.

Studies suggest alcohol can disrupt fetal gene regulation through epigenetic mechanisms (*Bestry et al., 2022*). DNA methylation is one epigenetic mechanism involving the catalytic addition of methyl groups to cytosine bases within cytosine-guanine (CpG) dinucleotide motifs during one-carbon metabolism. Methylation of DNA can alter chromatin density and influence patterns of gene expression in a tissue-specific and developmentally appropriate manner and disruption to this process may cause some of the difficulties experienced by people with FASD (*Jin and Liu, 2018*; *Fransquet et al., 2016*). Previous studies on human participants (*Lussier et al., 2018*; *Masemola et al., 2015*; *Jarmasz et al., 2019*; *Portales-Casamar et al., 2016*) and animals (*Chen et al., 2013*; *Abbott et al., 2018*; *Downing et al., 2011*) report that alcohol can disrupt DNA methylation either globally (*Jarmasz et al., 2019*; *Portales-Casamar et al., 2016*; *Chen et al., 2013*; *Abbott et al., 2018*) or at specific gene regions (*Masemola et al., 2015*; *Abbott et al., 2018*; *Downing et al., 2011*). Our recent systematic review, however, found limited replication of effects between studies suggesting the effects of alcohol on DNA methylation may be stochastic and influenced by numerous confounding factors (*Bestry et al., 2022*). PAE can either directly inhibit DNA methyltransferase enzymes or disrupt one-carbon metabolism via inhibition of bioavailability of dietary methyl donors, such as folate and choline to the fetus (*Chen et al., 2011*; *Hutson et al., 2012*). Choline, in particular, has been explored in several clinical trials to reduce cognitive deficits caused by PAE in affected individuals (*Nguyen et al., 2016*; *Wozniak et al., 2020*), or when administered during pregnancy (*Jacobson et al., 2018*; *Thomas et al., 2009*), with results suggesting a high methyl donor (HMD) diet could at least partially mitigate the adverse effects of PAE on various behavioural outcomes.

**eLife digest** Drinking excessive amounts of alcohol during pregnancy can cause foetal alcohol spectrum disorder and other conditions in children that affect their physical and mental development. Many countries advise women who are pregnant or trying to conceive to avoid drinking alcohol entirely. However, surveys of large groups of women in Western countries indicate that most women continue drinking low to moderate amounts of alcohol until they discover they are pregnant and then stop consuming alcohol for the rest of their pregnancy. It remains unclear how this common drinking pattern affects the foetus.

The instructions needed to build and maintain a human body are stored within molecules of DNA. Some regions of DNA called genes contain the instructions to make proteins, which perform many tasks in the body. Other so-called 'non-coding' regions do not code for any proteins but instead have roles in regulating gene activity. One way cells control which genes are switched on or off is adding or removing tags known as methyl groups to certain locations on DNA. Previous studies indicate that alcohol may affect how children develop by changing the patterns of methyl tags on DNA.

To investigate the effect of moderate drinking during the early stages of pregnancy, Bestry et al. exposed pregnant mice to alcohol and examined how this affected the patterns of methyl tags on DNA in their offspring. The experiments found moderate levels of alcohol were sufficient to alter the patterns of methyl tags in the brains and livers of the newborn mice. Most of the changes were observed in non-coding regions of DNA, suggesting alcohol may affect how large groups of genes are regulated. Fewer changes in the patterns of methyl tags were found in mice whose mothers had diets rich in two essential nutrients known as folate and choline.

Further experiments found that some of the affected mouse genes were similar to genes linked to foetal alcohol spectrum disorder and other related conditions in humans. These findings highlight the potential risks of consuming even moderate levels of alcohol during pregnancy and suggest that a maternal diet rich in folate and choline may help mitigate some of the harmful effects on the developing foetus.

Given the lack of clarity around the effects of typical patterns of alcohol consumption, which often do not cause observable phenotypes, we conducted an epigenome-wide association study of early moderate PAE in mice. Regions identified as sensitive to gestational alcohol exposure were replicated in human cohorts. The study was a controlled intervention investigating the impact of early moderate PAE on offspring DNA methylation comparing exposed and unexposed mice, with an additional arm comparing the effect of alcohol exposure in the context of an HMD maternal diet. The exposure period covers the equivalent of pre-conception up until the first trimester in humans when neurulation occurs, reflecting a typical situation in which women may consume alcohol up until pregnancy recognition (*Tsang et al., 2022*; *Muggli et al., 2016*). The primary outcome of the study was differences in offspring DNA methylation and secondary outcomes of behavioural deficits across neurodevelopmental domains relevant to FASD were also examined. We employed whole-genome bisulfite sequencing (WGBS) for unbiased assessment of CpG DNA methylation in newborn brain and liver, two target organs affected by ethanol (*Zakhari, 2006*), to explore tissue specificity of effects and to determine any 'tissue agnostic' effects which may have arisen prior to the germ layers separating in early gastrulation. We also conducted candidate gene testing of regions identified in prior studies as sensitive to early moderate PAE. Our study provides cogent evidence that common patterns of drinking can have measurable effects on fetal gene regulation, highlighting a role for maternal dietary support in public health interventions.

## Results
### Comparison of prenatal characteristics across treatment groups
To investigate the effects of early moderate PAE and an HMD diet across pregnancy on offspring DNA methylation and behavioural outcomes, we employed a murine model with four treatment groups. Treatment groups were designed to assess the effect of alcohol on offspring DNA methylation

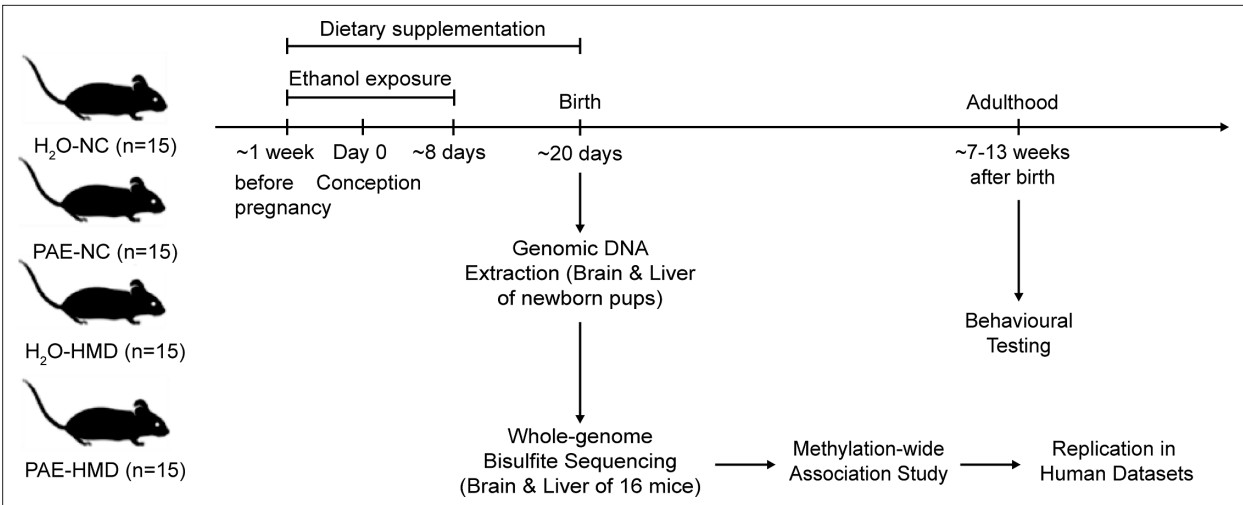

**Figure 1.** Overview of prenatal alcohol exposure (PAE) model. A schematic representation of the experiment design is shown in the figure. Fifteen dams were allocated to each treatment group. PAE mice were exposed to ethanol (10% vol/vol in non-acidified reverse osmosis drinking water ad libitum) from 1 week before pregnancy to gestational days 8–10 and the remaining mice received water ($H_2O$). The PAE and $H_2O$ groups received either normal chow (NC) or a high methyl donor (HMD) diet (NC containing 20 mg/kg folate and 4970 mg/kg choline) from 1 week before pregnancy until birth.

compared to control mice. An additional treatment group included a maternal diet high in methyl donors, with and without alcohol exposure (*Figure 1*).

The trajectory of weight gain during pregnancy was similar across all treatment groups with some evidence of more rapid weight gain in the HMD groups in the last 2 days (linear mixed effects regression model; $H_2O$-HMD: –2.282±0.918 g, p=0.0177; PAE-HMD: –1.656±0.814 g, p=0.0493; *Figure 2a*), although the total amount of weight gained between days 1 and 17–19 was not significantly different between treatment groups by linear mixed effects regression (*Figure 2a and b*). The total amount of liquid consumed over the course of pregnancy was significantly lower in HMD dams by unpaired t-test (*Figure 2b* and *Figure 3c*). There was no significant difference in the average litter size (6.525±0.297 pups, *Figure 2c*) and pup sex ratios (*Figure 2d*) between treatment groups by unpaired t-test.

## Effects of early moderate PAE on DNA methylation in newborn mice tissue

A subset of 16 newborn pups (n=4 per treatment) matched for sex and litter size were selected for WGBS. Both neonatal brain and liver were harvested to investigate the effects of early moderate PAE on fetal CpG DNA methylation. A total of 21,842,961 CpG sites were initially available for analysis.

Global levels of DNA methylation stratified across different genomic contexts were preserved across treatment conditions, with no major differences in average DNA methylation content between groups (*Figure 4*). To investigate region-specific effects of early moderate PAE on newborn DNA methylation, we conducted genome-wide testing comparing exposed and unexposed mice on the normal chow. We identified 78 differentially methylated regions (DMRs) in the brain and 759 DMRs in the liver (p<0.05 and mean difference in methylation across the DMR with PAE (delta) >0.05) from ~19,000,000 CpG sites tested after coverage filtering (*Figure 5a and b*). These regions were annotated to nearby genes using *annotatr* and are provided in *Supplementary file 1a and b*. Two of the DMRs overlapped in mouse brain and liver (tissue agnostic), but the remainder were tissue specific. Among these tissue agnostic regions was the *Impact* gene on chromosome 18, which had lower methylation in PAE+NC mice compared to $H_2O$+NC mice in both the brain and liver (*Figure 5c and d*). The other tissue agnostic region was within 5 kb downstream of *Bmf* and had higher DNA methylation in brain and liver tissue of PAE+NC mice.

Lower DNA methylation with early moderate PAE in NC mice was more frequently observed in liver DMRs (93.5% of liver DMRs), while brain DMRs were almost equally divided between lower and higher DNA methylation with early moderate PAE (52.6% of brain DMRs had lower DNA methylation with

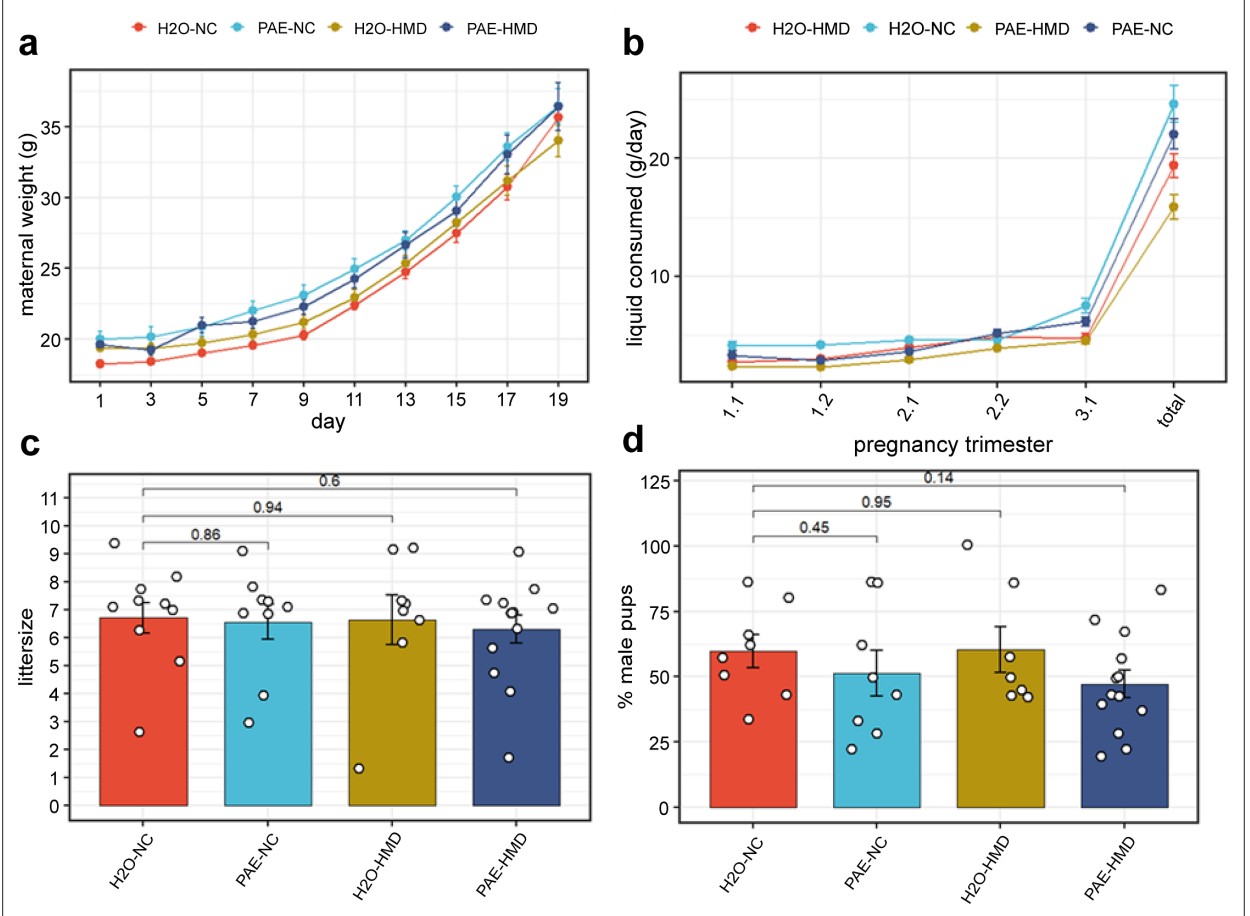

**Figure 2.** PAE and HMD effects on dam characteristics. (**a**) Dam weight progression was significantly affected by HMD but not PAE by quadratic mixed effects model without interaction. (**b**) Trajectory of liquid consumption across pregnancy was affected by PAE and HMD by quadratic mixed effects model. PAE and HMD significantly interacted with trimester of pregnancy. (**c**) Litter size (N=40) and (**d**) pup sex ratios (N=36) were not significantly associated with PAE or HMD by unpaired t-test or ANOVA. All line and bar plots show mean and standard deviation. NC = normal chow, HMD = high methyl diet, PAE = prenatal alcohol exposure. Comparisons show p-value by unpaired t-test compared to the $H_2O$-NC baseline treatment group.

early moderate PAE). Some DMRs localised to the same genes in both brain and liver, although they were different regions. The three genes affected by PAE in both brain and the liver tissues were the Autism Susceptibility Gene 2 (*Auts2*), Androglobin (*Adgb*), and RNA Binding Protein Fox 1 (*Rbfox1*) genes (*Table 1*). In both brain and liver tissues, DMRs were enriched in non-coding intergenic and open sea regions and relatively underrepresented in coding and CpG island regions (*Figure 5e and f*). Using open chromatin assay and histone modification datasets from the ENCODE project, we found enrichment ($p<0.05$) of DMRs in open chromatin regions (ATAC-seq), enhancer regions (H3K4me1), and active gene promoter regions (H3K27ac), in mouse fetal forebrain tissue and fetal liver (*Table 2*). Gene ontology enrichment analysis of liver DMRs that did localise to genes showed enrichment in 10 predominantly neuronal pathways, with neuron projection being the most significant (*Figure 5g*, *Supplementary file 1c and d*).

## HMD mitigates the effects of early moderate PAE on DNA methylation

To determine whether administration of an HMD throughout pregnancy could mitigate the effects of PAE on offspring DNA methylation, we examined alcohol-sensitive DMRs identified in the previous analysis in the HMD mice. Compared to control mice ($H_2O$+NC), PAE+HMD mice exhibited significant ($p<0.05$) DNA methylation differences in only 12/78 (15%) brain (*Supplementary file 1g*), and 124/759 (16%) liver (*Supplementary file 1h*) DMRs, suggesting the effects were predominantly mitigated. Effect sizes compared to mice on the normal chow were substantially lower, in some cases more than 25% reduced in mice on the HMD diet (*Figure 6*).

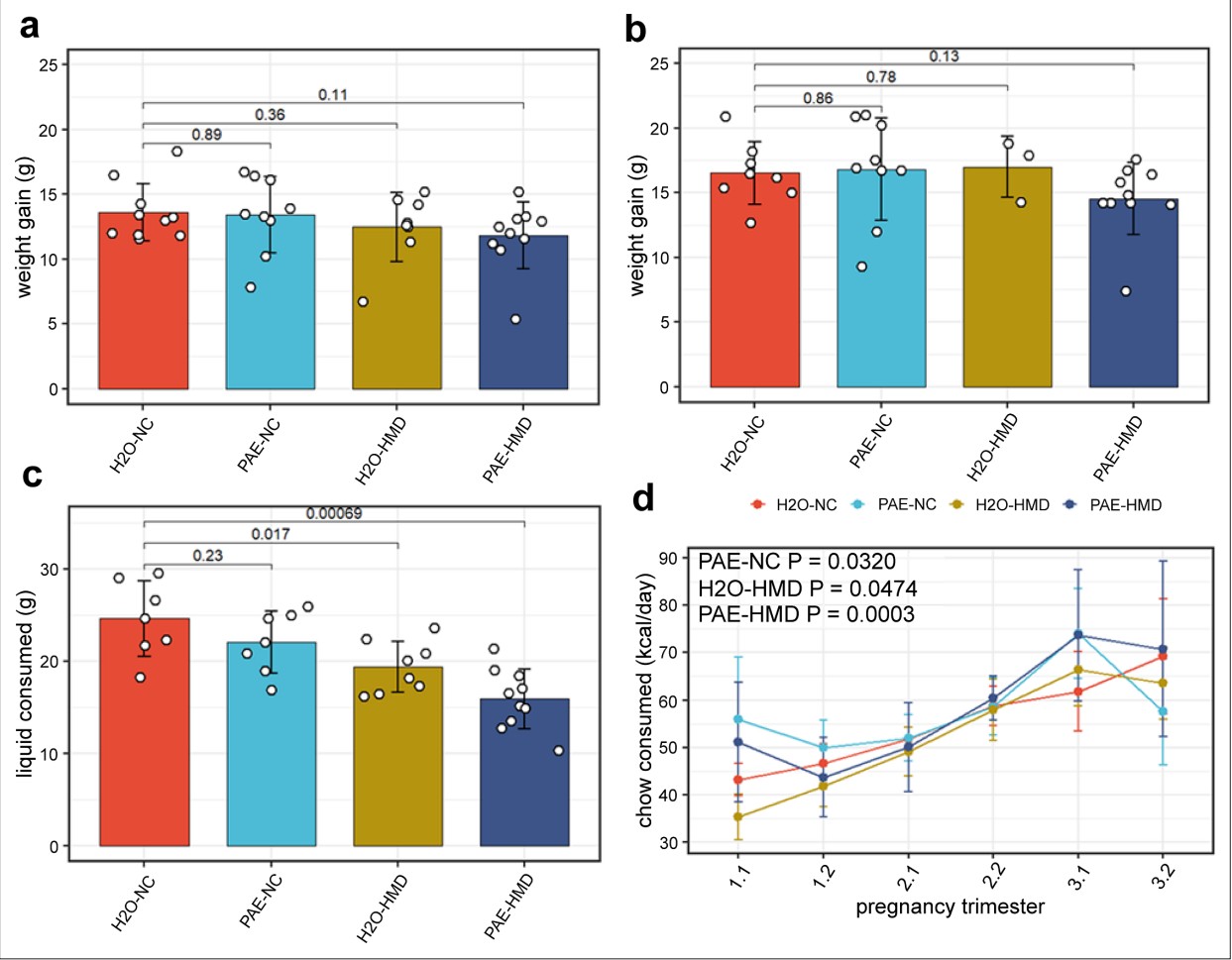

**Figure 3.** Prenatal alcohol exposure (PAE) and high methyl donor (HMD) effects on dam characteristics. There was no significant difference in the average gain of weight in dams between (**a**) days 1–17 and (**b**) days 1–19 by treatment group. Both timepoints were included due to some pregnancies ending by day 19. (**c**) Dams given supplemented chow consumed significantly lower total quantity of liquid across pregnancy. Bar plots show mean and standard deviation for each treatment group. Each point represents one dam. (**d**) The trajectory of chow consumed by dams across pregnancy significantly varied with the addition of treatments. Points show mean and standard deviation for each treatment group. Statistical analysis involved linear mixed effects regression comparing trajectories of treatment groups to H₂O-NC baseline control group. N=36.

## Effects of early moderate PAE and HMD on behavioural outcomes in adult mice

Remaining littermates from each treatment group were reared to adulthood and underwent behavioural testing assessing various neurocognitive domains that can be affected in FASD including locomotor activity, anxiety, spatial recognition, memory, motor coordination, and balance. There was no evidence that early moderate PAE had a significant effect on any of the behavioural outcomes tested (*Figure 7*). Mice exposed to HMD exhibited greater locomotor activity, in terms of distance travelled (*Figure 8*).

## Replication studies in human PAE and FASD case-control cohorts

We undertook validation studies by examining PAE-sensitive regions identified in our murine model using existing DNA methylation data from human cohorts to address the generalizability of our findings. Only 36 of the 78 (46.2%) brain DMRs, and 294 of the 759 (38.8%) liver DMRs, had homologous regions in the human genome that were able to be tested. In this validation study, DNA methylation array data from 147 newborn buccal swabs from the Asking Questions About Alcohol in Pregnancy (AQUA) cohort (*Muggli et al., 2022*) were available from this cohort (96 moderate PAE and 51 controls). We performed differential testing on a total of 1898 CpG sites that corresponded to

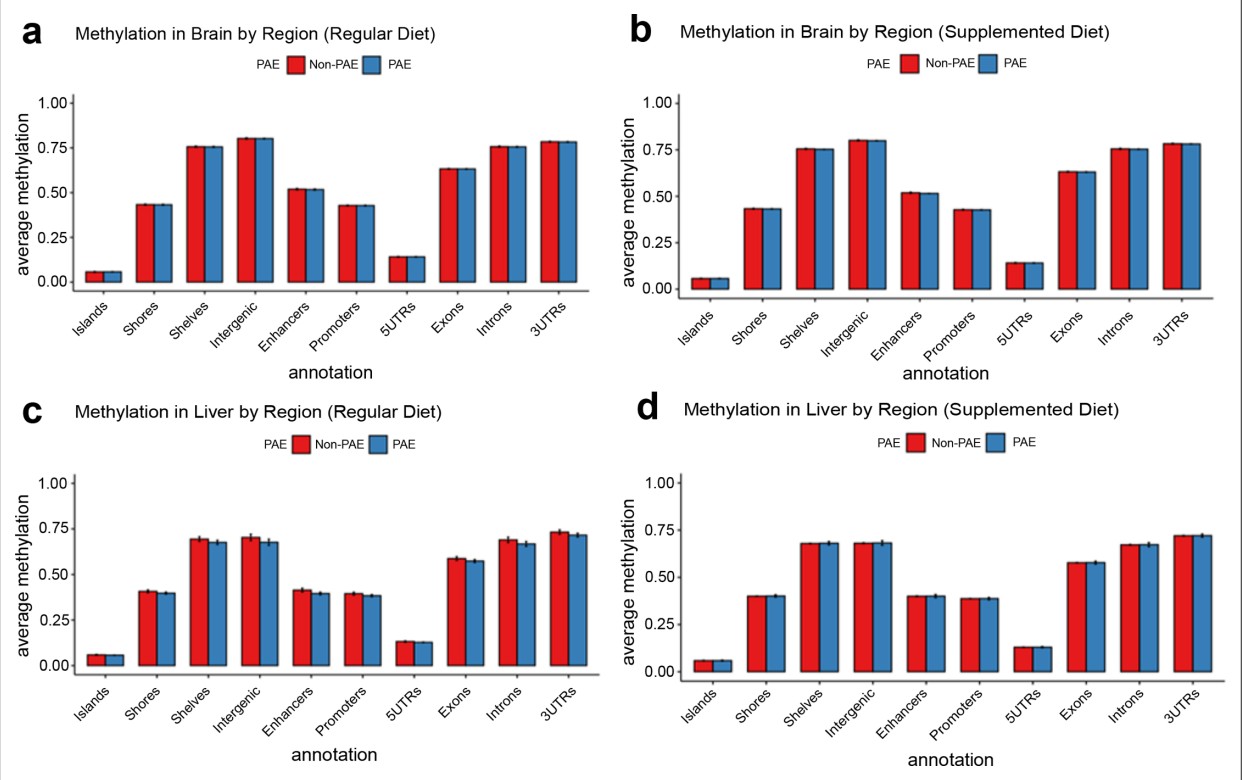

**Figure 4.** No evidence for global disruption of methylation by prenatal alcohol exposure (PAE). The figure shows methylation levels averaged across cytosine-guanines (CpGs) in different regulatory genomic contexts. Neither brain tissue (**a** and **b**) nor liver tissue (**c** and **d**) were grossly affected by PAE exposure (blue bars). Bars represent means and standard deviation.

mouse DMRs, comparing 'never exposed' newborns to 'any exposure' and found no evidence of differential DNA methylation at these CpG (data not shown). We also accessed publicly available DNA methylation array measurements from buccal swabs taken from a Canadian clinical cohort of children with diagnosed FASD and controls (GSE109042). To avoid confounding due to ancestry, we analysed the 118 Caucasian individuals (30 FASD and 88 controls). Differential testing of a total of 2316 CpG sites that were homologous to mouse DMRs statistically replicated seven DMR associations with FASD status (FDR p<0.05) after adjusting for participant age, sex, array number, and estimated cell counts (*Table 3*). Visual comparisons of DNA methylation across these seven DMRs revealed striking differences in effect sizes between people with FASD and our murine model (*Figure 9*). Genes associated with these DMRs are linked to clinically relevant traits in the GWAS catalogue including facial morphology (*GADD45A*; *Indencleef et al., 2021*), educational attainment (*AP2B1*; *Okbay et al., 2022*), intelligence (*RP9*; *Davies et al., 2019*), autism and schizophrenia (*ZNF823*; *Autism Spectrum Disorders Working Group of The Psychiatric Genomics Consortium, 2017*).

## Candidate gene analysis of previously defined alcohol-sensitive regions

We also undertook a replication analysis in our murine data of previously published alcohol-sensitive regions by undertaking a systematic review of previously published mammalian studies (*Bestry et al., 2022*). Candidate gene studies identified 21 CpG sites (FDR<0.05) in the brain from 15,132 CpG sites tested, including two sites in the *Mest* (*Peg1*) gene and 19 sites in *Kcnq1* (*KvDMR1*) (*Supplementary file 1i*). There were nine FDR-significant CpG sites identified in the liver out of 15,382 CpG sites tested, all of which were in *Peg3* (*Supplementary file 1j*). All FDR-significant CpG sites from both tissues had higher DNA methylation in mice with PAE.

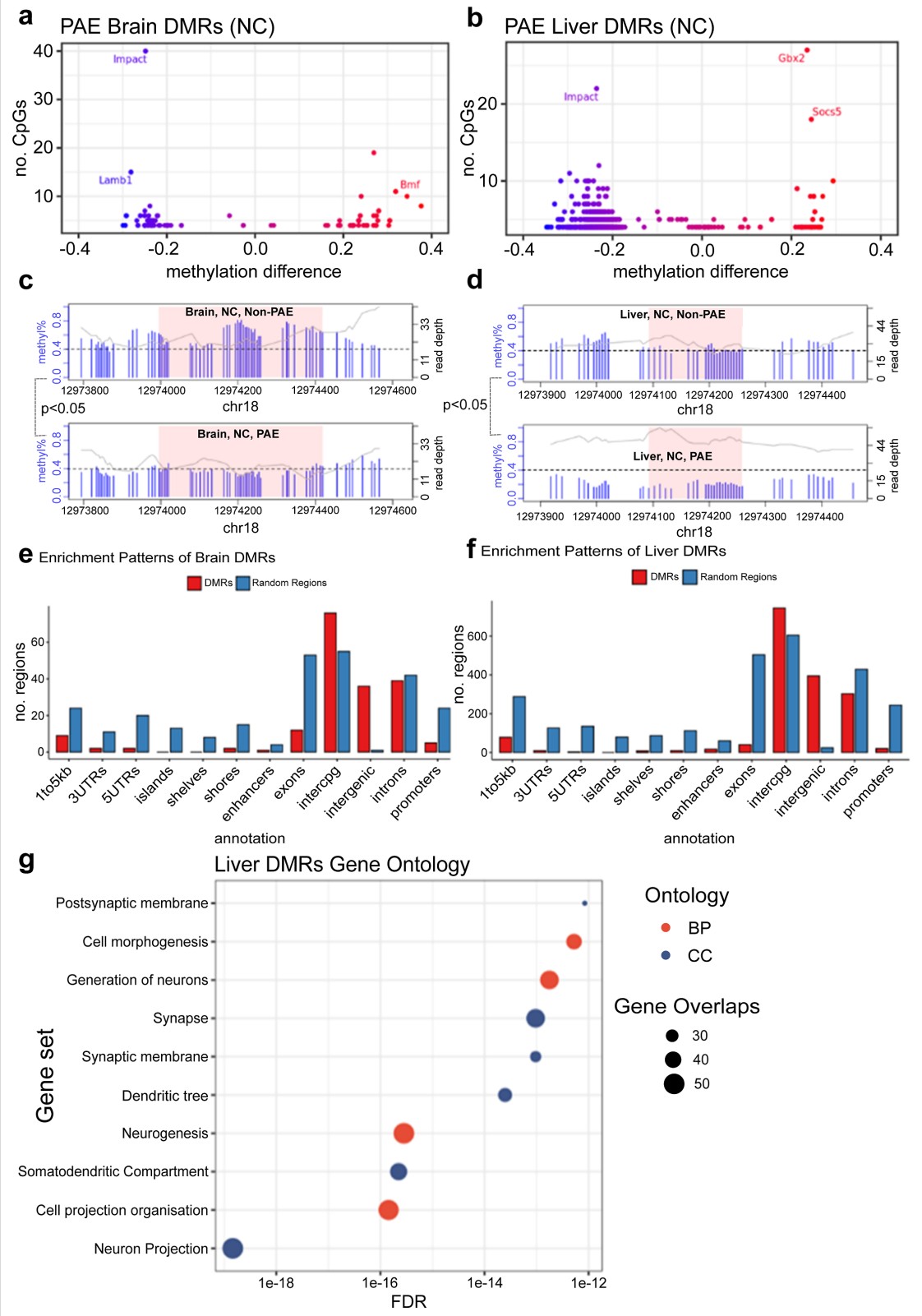

**Figure 5.** Prenatal alcohol exposure (PAE) was associated with site-specific differences in offspring DNA methylation. The majority of differentially methylated regions (DMRs) lost methylation with PAE in (**a**) brain and (**b**) liver of mice given normal chow (NC). Each point represents one DMR. Point colour indicates change in DNA methylation with PAE. PAE was also associated with lower methylation in the DMRs identified in the promoter of the *Impact* gene in (**c**) brain and (**d**) liver, within NC mice. Each plot represents a separate treatment group. Each blue vertical line indicates a cytosine-

*Figure 5 continued on next page*

*Figure 5 continued*

guanine (CpG) site, with the height and corresponding left y-axis indicating the methylation ratio. The grey line and corresponding right y-axis indicate coverage at each CpG site. The black horizontal dotted line indicates 40% methylation for comparison purposes. The x-axis indicates the base position on chromosome 18, with the pink shaded area highlighting the DMR. DMR plots include 200 base pair flanking regions on each side of the DMR. DMRs identified in (**e**) brain and (**f**) liver were enriched in intergenic and inter-CpG regions, whilst being underrepresented in CpG and gene regions. The bar plot compares the number of whole-genome bisulfite sequencing (WGBS) DMRs in red to a set of equivalent randomly generated regions in blue. (**g**) Gene ontology analysis of liver DMRs shows enrichment within neuronal cellular components and biological processes. BP/red point = biological process, CC/blue point = cellular component. X-axis of point indicates FDR of ontology. Size of point indicates number of overlapping genes with ontology. There were insufficient number of DMRs identified in the brain for a gene ontology analysis.

## Discussion

In this study, we found that moderate early (first trimester) PAE was sufficient to induce site-specific differences to DNA methylation in newborn pups without causing overt behavioural outcomes in adult mice. Although global levels of DNA methylation were not significantly different with PAE, regional analysis demonstrated widespread effects characterised predominantly by lower DNA methylation with PAE, mostly at non-coding regions of the genome. In our model, alcohol effects on DNA methylation were predominantly tissue-specific, with only two genomic regions and four genes that were similarly affected in both liver and brain. These perturbations may have been established stochastically because of PAE to the early embryo and maintained in the differentiating tissue. Further analysis in different germ layer tissues is required to formally establish this. Indeed, most of the observed effects were tissue-specific, with more perturbations to the epigenome observable in liver tissue, which may reflect the liver's specific role in metabolic detoxification of alcohol. Alternatively, cell-type composition differences between brain and liver might explain differential sensitivity to alcohol effects. Generally, DMRs were enriched in non-coding regions of the genome with regulatory potential, suggesting alcohol has broad effects on genome regulation.

Both the human replication studies and the candidate gene analysis provide validity to our model for recapitulating some of the genomic disturbances reported in patients with clinical FASD. It is remarkable that some associations identified in our murine model of early moderate exposure were recapitulated in human subjects with FASD despite species and biosample differences, suggesting that at least some DNA methylation changes are stable over time. Notably the effect size for replicated regions were strikingly smaller in blood samples from subjects with FASD suggesting the dose and duration of exposure may need to exceed a high threshold to survive reprogramming in the blood. We speculate this may explain lack of reproducibility in the AQUA cohort.

In the candidate gene analysis we replicated previously published reports of decreased DNA methylation within *Peg3* and *KvDMR1* from South African children with FAS (***Masemola et al., 2015***). Both genes are methylated in a parent-of-origin specific manner, suggesting that alcohol may affect imprinting processes. Previous rodent and human studies have identified DNA methylation differences with PAE in imprinted regions such as the *Igf2/H19* locus (***Portales-Casamar et al., 2016***; ***Downing et al., 2011***; ***Zhou et al., 2016***), although results are not entirely consistent (***Marjonen et al., 2018***; ***Stouder et al., 2011***). On the balance of this, we speculate duration of exposure, dose,

**Table 1.** Table of differentially methylated regions (DMRs) identified in the intronic regions of genes that contained DMRs in both the brain and liver.
Δmeth indicates the percentage change in average methylation level within the DMR with prenatal alcohol exposure (PAE) compared to non-PAE mice.

| Gene | Tissue | Intronic DMR | Width | No. CpGs | Δmeth | p-Value |
|------|--------|--------------|-------|----------|-------|---------|
| *Auts2* | Brain | chr5:131510296–131510465 | 170 | 5 | –23.5% | <0.05 |
| *Auts2* | Liver | chr5:131621828–131621999 | 172 | 4 | –22.5% | <0.05 |
| *Adgb* | Brain | chr10:10455557–10455883 | 327 | 4 | –25.0% | <0.05 |
| *Adgb* | Liver | chr10:10353338–10353613 | 276 | 4 | –25.9% | <0.05 |
| *Rbfox1* | Brain | chr16:6813039–6813217 | 179 | 5 | –24.3% | <0.05 |
| *Rbfox1* | Liver | chr16:6781985–6782330 | 346 | 5 | –22.6% | <0.05 |

**Table 2.** Number and percentage of brain and liver differentially methylated regions (DMRs) that overlap with tissue-specific regulatory regions.

ATAC-seq, H3K4me1, and H3K27ac regions were obtained at 0 days postnatal from the ENCODE database. p-Values for permutation testing using a randomisation strategy.

| Assay type | Brain DMRs | Brain randomised regions | Liver DMRs | Liver randomised regions |
|---|---|---|---|---|
| ATAC-seq | 21/78 (26.92%), p=0.01 | 1/78 (1.28%), p=0.16 | 53/759 (6.98%) p=0.01 | 22/759 (2.90%) p=0.31 |
| H3K4me1 | 4/78 (5.13%) p=0.03 | 2/78 (2.56%) p=0.18 | 38/759 (5.01%) p=0.05 | 35/759 (4.61%) p=0.32 |
| H3K27ac | 9/78 (11.54%) p=0.01 | 2/78 (2.56%) p=0.74 | 48/759 (6.32%) p=0.01 | 19/759 (2.50%) p=0.26 |

and other tissue-related factors all likely influence the extent to which genome regulation is perturbed and manifests as differences in DNA methylation.

Our results are encouraging for biomarker studies and aid in the prioritisation of associations for future follow-up, particularly in relation to diagnosis of FASD. For example, three genes that were validated in the *Lussier et al., 2018* cohort are zinc finger proteins (*RP9*, *PEX12*, and *ZNF823*) that play an important role in fetal gene regulation. Notably, *PEX12* is associated with Zellweger syndrome, which is a rare peroxisome biogenesis disorder (the most severe variant of peroxisome biogenesis disorder spectrum), characterised by neuronal migration defects in the brain, dysmorphic craniofacial features, profound hypotonia, neonatal seizures, and liver dysfunction (*Konkol'ová et al., 2015*).

Future studies could perform transcriptomic analysis to investigate. Another key finding from this study was that HMD mitigated some of the effects of PAE on DNA methylation. Although a plausible alternative explanation is that some of the PAE regions were not reproduced in the set of mice given the folate diet, our data are consistent with preclinical studies of choline supplementation in rodent models (*Thomas et al., 2007*; *Thomas et al., 2000*; *Otero et al., 2012*). Moreover, a subset of PAE regions were statistically replicated in subjects with FASD, suggestive of robust associations. Although our findings should be interpreted with caution, they collectively support the notion that alcohol-induced perturbation of epigenetic regulation may occur, at least in part, through disruption of the one-carbon metabolism. The most encouraging aspect of this relates to the potential utility for evidence-informed recommendations for dietary advice or supplementation, particularly in population groups with limited access to antenatal care or healthy food choices.

Strengths of this study include the use of controlled interventions coupled with comprehensive assessment of the effects of PAE on multiple tissues. We also performed WGBS representing the gold standard in DNA methylation analysis, which to our knowledge has not been performed before in the context of murine PAE studies. Our findings were partially generalisable in replication studies addressing the robustness of our experimental approach. Caveats of our study design include a limited ability to determine the contribution of specific cell types within tissues to the methylation differences observed, and we did not assess markers of brain or liver physiology. Additionally, we employed an ad libitum alcohol exposure model rather than direct dosing of dams. Although the trajectories of alcohol consumption were not statistically different between groups, this introduces more variability into alcohol exposure patterns, and might impact offspring methylation data. Despite these limitations, the results were meaningful in the context of typical patterns of alcohol consumption in human populations.

In conclusion, this study demonstrates that early moderate PAE can disturb fetal genome regulation in mice and humans and supports current public health advice that alcohol consumption during pregnancy, even at low doses, may be harmful.

## Materials and methods
### Murine subjects and housing
To study the effects of early moderate PAE on offspring DNA methylation processes, we adapted a murine model study design that has previously reported DNA methylation changes at the A$^{vy}$ locus in Agouti mice (*Kaminen-Ahola et al., 2010*; *Figure 1*). This study received animal ethics approval

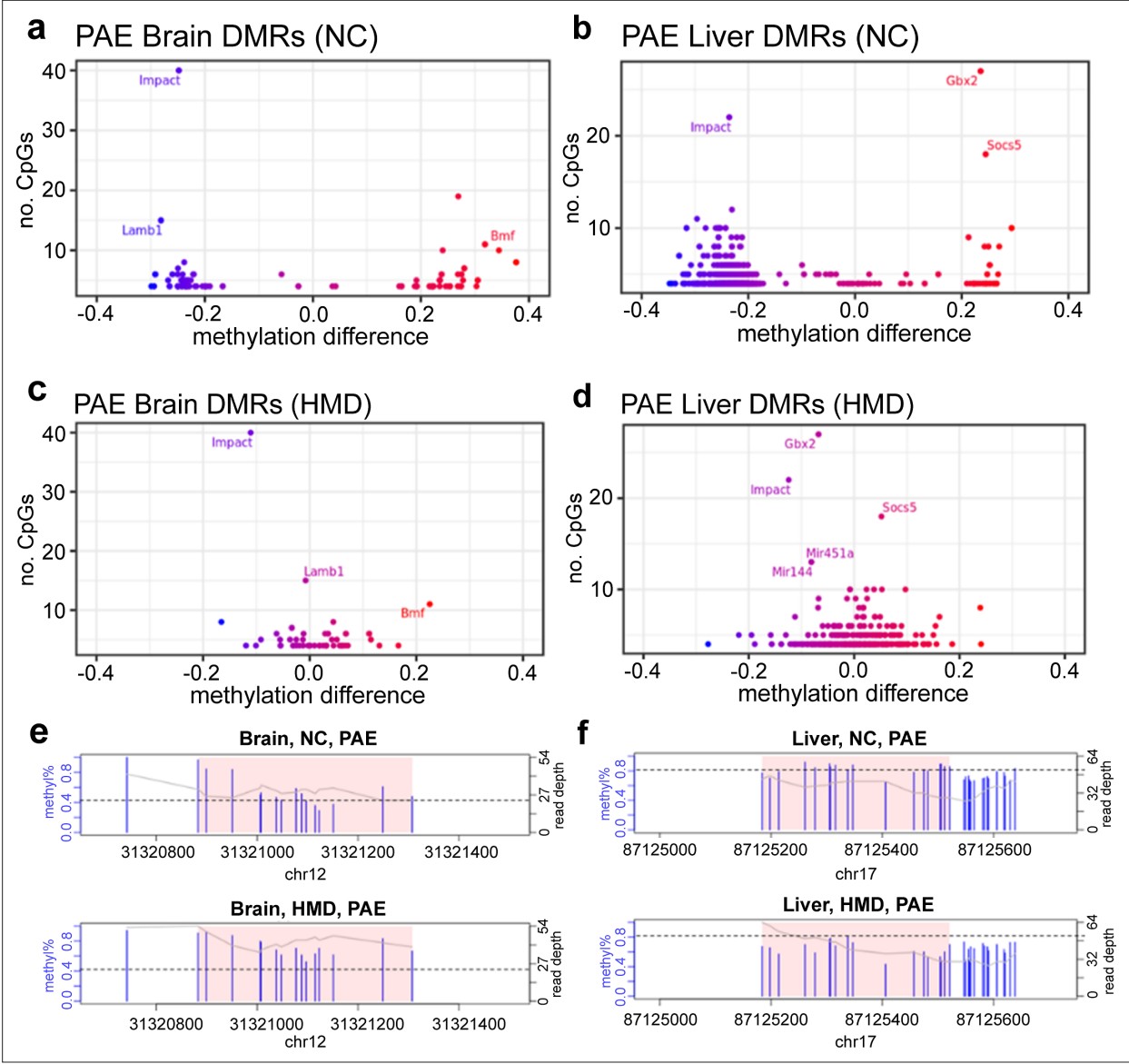

**Figure 6.** High methyl donor (HMD) partially mitigated effects of prenatal alcohol exposure (PAE) on offspring DNA methylation. Average DNA methylation effect sizes above 30% with PAE were observed in some (**a**) brain and (**b**) liver differentially methylated regions (DMRs) in normal chow (NC) mice. Mean absolute difference in methylation with PAE is reduced within the HMD mice in (**c**) brain and (**d**) liver. Each point represents one DMR. Point colour indicates change in DNA methylation with PAE. Points with a high number of cytosine-guanines (CpGs) and methylation difference are annotated with associated gene if located within a genic region. HMD was associated with (**e**) higher methylation in the DMR identified proximal to *Lamb1* on chromosome 12 in brain and (**f**) lower methylation in the DMR identified proximal to *Socs5* on chromosome 17 in liver. Each plot represents a separate treatment group. Each blue vertical line indicates a CpG site, with the height and corresponding left y-axis indicating the methylation ratio. The grey line and corresponding right y-axis indicate coverage at each CpG site. The black horizontal line indicates (**e**) 40% and (**f**) 80% methylation for comparison purposes. The x-axis indicates the base position on the chromosome, with the pink shaded area highlighting the DMR. DMR plots include 200 base pair flanking regions on each side of the DMR.

from the Telethon Kids Institute Animal Ethics Committee (Approval Number: 344). Sixty nulliparous C57BL/6J female mice aged ~8 weeks were mated with equivalent stud male mice. Pregnant dams were randomly assigned to one of four treatment groups (n=15 dams per group) that varied based on composition of the drinking water and chow given to the dams:

i. PAE-NC (prenatal alcohol exposure-normal chow): 10% (vol/vol) ethanol in non-acidified water ad libitum from 10 days prior to mating until gestational days (GD) 8–10. This is intended to replicate typical patterns of drinking during the first trimester of pregnancy in humans. Dams

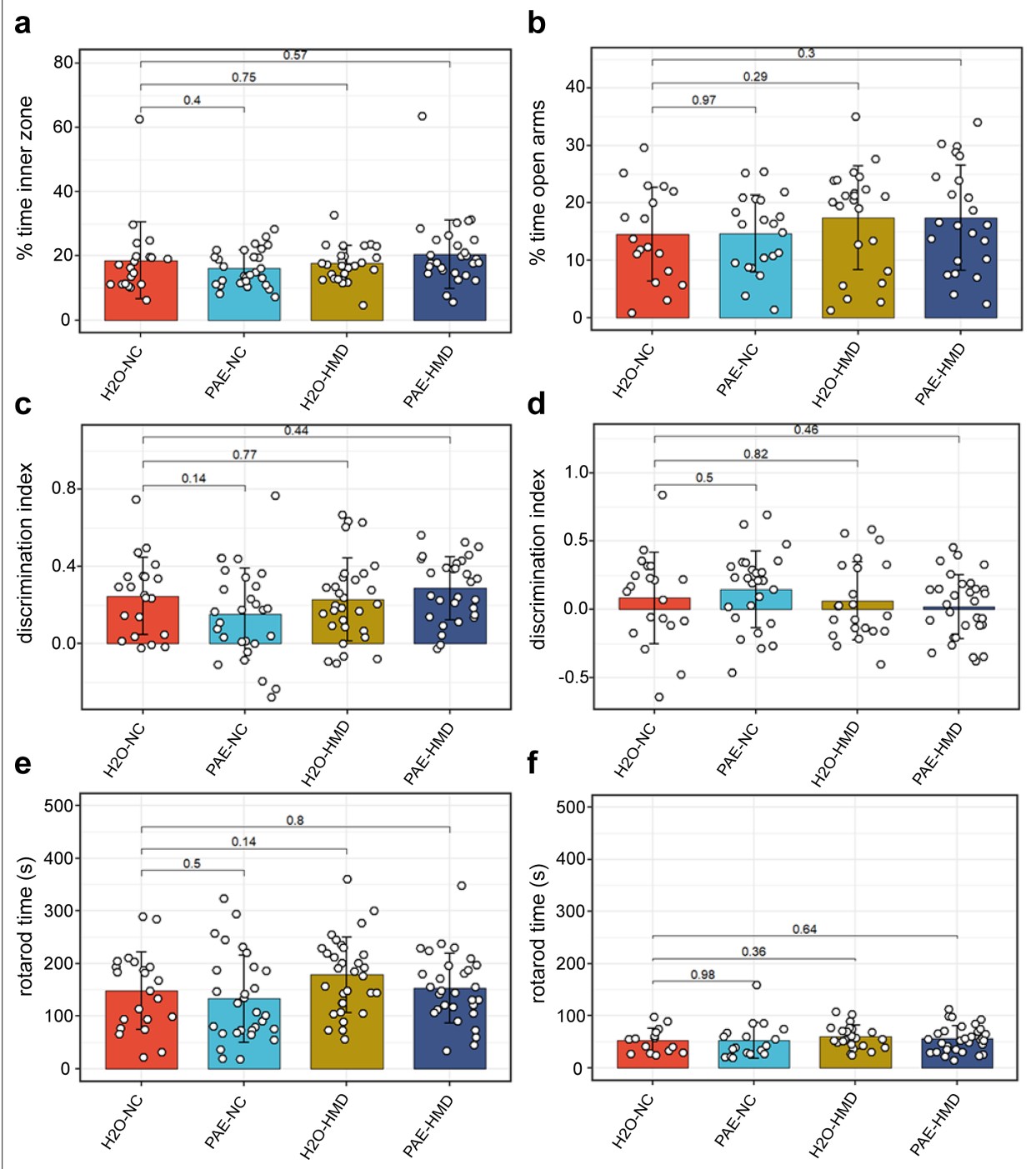

**Figure 7.** PAE had no significant effect on other assessed behavioural outcomes. PAE and HMD had no significant effect on anxiety as evident by no significant difference by unpaired t-test in the (**a**) percent time in the inner zone in the open field test (N=104) and (**b**) percent time open arms in the elevated plus maze test (N=85). PAE and HMD had no significant effect on spatial recognition as evident by no significant difference by unpaired t-test in the discrimination index in (**c**) object recognition (N=108) and (**d**) object in place test (N=98). PAE and HMD had no significant effect on motor coordination and balance as evident by no significant difference by unpaired t-test in times in (**e**) first rotarod test (N=112) and (**f**) second rotarod test (N=87). Bars show mean and standard deviation. Each point represents one mouse. NC = normal chow, HMD = high methyl diet, PAE = prenatal alcohol exposure. Time interval for each mouse was (**a–c**) 300 s and (**d**) 180 s.

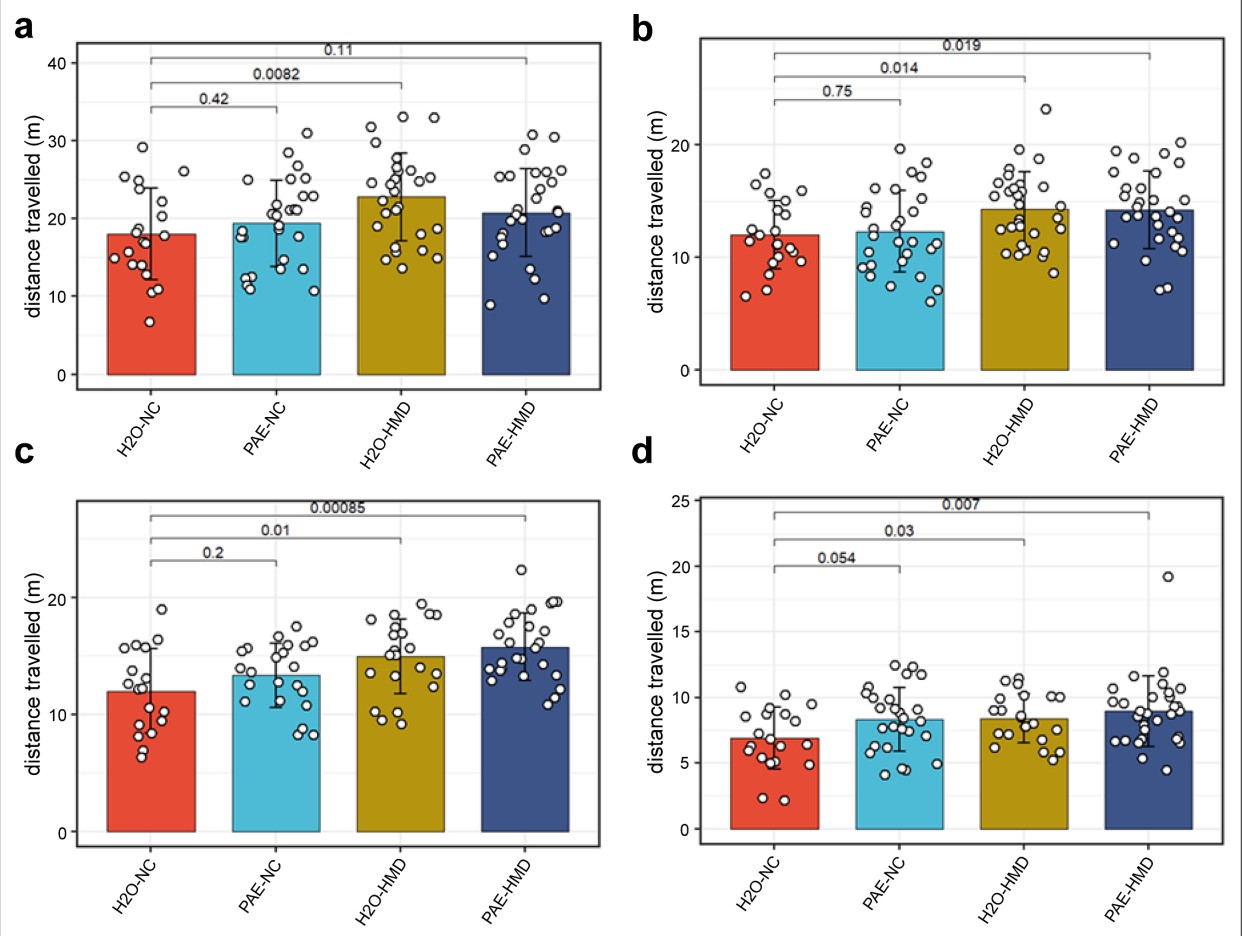

**Figure 8.** HMD was associated with increased locomotor activity. HMD was associated with increased locomotor activity compared to NC, indicated by significantly greater total distance travelled in the (**a**) open field test (N=104), (**b**) object recognition test (N=108), (**c**) elevated plus maze test (N=88), and (**d**) object in place test (N=98) by unpaired t-test. Bars show mean and standard deviation. Each point represents one mouse. NC = normal chow, HMD = high methyl diet, PAE = prenatal alcohol exposure. Time interval for each mouse was (**a–c**) 300 s and (**d**) 180 s.

received non-acidified reverse osmosis water for the remainder of pregnancy and normal chow (Rat and Mouse Cubes, Speciality Feeds, Glen Forrest, Australia) throughout pregnancy.

ii.  PAE-HMD (prenatal alcohol exposure-high methyl donor diet): 10% (vol/vol) ethanol in non-acidified water ad libitum from 10 days prior to mating until GD8-10 and non-acidified reverse osmosis water for remainder of pregnancy. Isocaloric HMD chow consisting of 20 mg/kg folate and 4970 mg/kg choline throughout pregnancy (Speciality Feeds, Glen Forrest, Australia).

iii.  $H_2O$-NC (water-normal chow): non-acidified water and normal chow throughout pregnancy.

iv.  $H_2O$-HMD (water-high methyl donor diet): non-acidified water and HMD chow throughout pregnancy.

## WGBS of newborn mouse tissues

Pups selected for WGBS in each intervention group were matched on sex and litter size to minimize variability in exposure. Two male and two female pups per treatment group (n=16 total) were euthanised by intraperitoneal injection with ketamine and xylazine on the day of birth for WGBS of their brain and liver tissues. Mouse tissue samples were stored at –80°C. Remaining littermates grew until adulthood for behavioural testing. Ten milligrams of tissue were collected from each liver and brain and lysed in Chemagic RNA Tissue10 Kit special H96 extraction buffer. Total nucleic acid was extracted from the tissues using the Chemagic 360 instrument (PerkinElmer) and quantified with Qubit DNA High Sensitivity Kit (Catalogue Number: Q32854, Thermo Scientific). 100 ng of genomic DNA was spiked with 0.5 ng of unmethylated lambda DNA (Catalogue Number: D1521, Promega)

**Table 3.** Differentially methylated regions (DMRs) identified in the murine model that were validated in the **Lussier et al., 2018** human case-control cohort for a clinical diagnosis of fetal alcohol spectrum disorder (FASD).
The upper section describes properties of **Lussier et al., 2018** human DMRs. The lower section describes properties of the equivalent murine model DMRs.

| DMR | Organism | Tissue | Chr | Start | End | Width | No. CpGs | FDR | Meandiff | Gene |
|---|---|---|---|---|---|---|---|---|---|---|
| 1 | Human | Buccal | 1 | 68151571 | 68152310 | 740 | 5 | 0.028636 | –0.00497 | *GADD45A* |
| 2 | Human | Buccal | 19 | 13000782 | 13002357 | 1576 | 11 | 0.000197 | –0.00203 | *GCDH* |
| 3 | Human | Buccal | 7 | 33148815 | 33149316 | 502 | 11 | 0.001149 | –0.00011 | *RP9* |
| 4 | Human | Buccal | 17 | 33905444 | 33905888 | 445 | 14 | 0.000171 | –0.00359 | *AP2B1, PEX12* |
| 5 | Human | Buccal | 17 | 27181503 | 27182342 | 840 | 11 | 0.018536 | –0.00246 | *ERAL1, FAM222B* |
| 6 | Human | Buccal | 19 | 12992181 | 12992479 | 299 | 9 | 0.037431 | –0.00179 | *CTD-2265O21.7, DNASE2* |
| 7 | Human | Buccal | 19 | 11849531 | 11850013 | 483 | 9 | 0.022724 | –0.00244 | *ZNF823* |
| 1 | Mouse | Liver | 6 | 67034885 | 67035082 | 197 | 4 | <0.05 | –0.220833 | *E230016M11Rik* |
| 2 | Mouse | Liver | 8 | 84901298 | 84901544 | 246 | 5 | <0.05 | –0.234457 | *Klf1* |
| 3 | Mouse | Liver | 9 | 22453836 | 22453893 | 57 | 5 | <0.05 | –0.226427 | *Rp9* |
| 4 | Mouse | Brain | 14 | 21403570 | 21403622 | 52 | 4 | <0.05 | –0.234193 | *Adk* |
| 5 | Mouse | Liver | 11 | 78069463 | 78070002 | 539 | 9 | <0.05 | –0.255864 | *Mir144, Mir451a* |
| 6 | Mouse | Liver | 11 | 78072079 | 78072313 | 234 | 4 | <0.05 | –0.215227 | *Mir144, Mir451a* |
| 7 | Mouse | Liver | 2 | 177091927 | 177092945 | 1018 | 5 | <0.05 | –0.224354 | Intergenic |

to assess the bisulfite conversion efficiency. Each sample was digested with 2 µl RNase A (Invitrogen) at 37°C for 20 min to remove RNA. 100 ng of genomic DNA from each sample was sheared using a Covaris M220 (300 bp settings, Covaris). Libraries were prepared using the Lucigen NxSeq AmpFREE Low DNA Library Kit (Catalogue Number: 14000-1, Lucigen), according to the manufacturer's instructions. Nextflex bisulfite-seq barcodes (Catalogue Number: Nova-511913, PerkinElmer) were used as the adapters with incubation at 25°C for 30 min. The libraries were bisulfite converted using the Zymo EZ DNA Methylation-Gold Kit (Catalogue Number: D5005, Zymo Research) and PCR amplified using the KAPA HiFi Uracil PCR Kit (Catalogue Number: ROC-07959052001, Kapa Biosystems). The final libraries were assessed with the Agilent 2200 Tapestation System using D1000 Kit (Catalogue Number: 5067-5582). WGBS was performed by Genomics WA sequencing core on a NovaSeq 6000 (Illumina) using 2×150 bp chemistry on an S4 flow cell. The bisulfite conversion rate in each tissue sample was at least 99%. The overall mean coverage in each sample was 9.69× (range: 6.51–12.12×).

## Behavioural testing in adult mice

Littermates who were not sacrificed at birth were reared on normal chow and drinking water ad libitum until adulthood (~8 weeks after birth) when they underwent behavioural tests assessing five neurodevelopmental domains that can be affected by PAE including locomotor activity, anxiety, spatial recognition, memory, motor coordination, and balance. These tests included the open field test (locomotor activity, anxiety) (**Seibenhener and Wooten, 2015**), object recognition test (locomotor activity, spatial recognition) (**Lueptow, 2017**), object in place test (locomotor activity, spatial recognition) (**Murai et al., 2007**), elevated plus maze test (locomotor activity, anxiety) (**Komada et al., 2008**), and two trials of the rotarod test (motor coordination, balance) (**Deacon, 2013**). Between mouse subjects, behavioural testing equipment was cleaned with 70% ethanol. Video recording was employed for all behavioural tests, except for the rotarod, and the assessment process was semi-automated using ANY-maze software (Stoelting Co., Wood Dale, IL, USA).

## Statistical analysis

Dam characteristics and pup behavioural testing results were generally assessed using unpaired t-tests comparing each treatment group to the baseline control group that was given non-acidified reverse

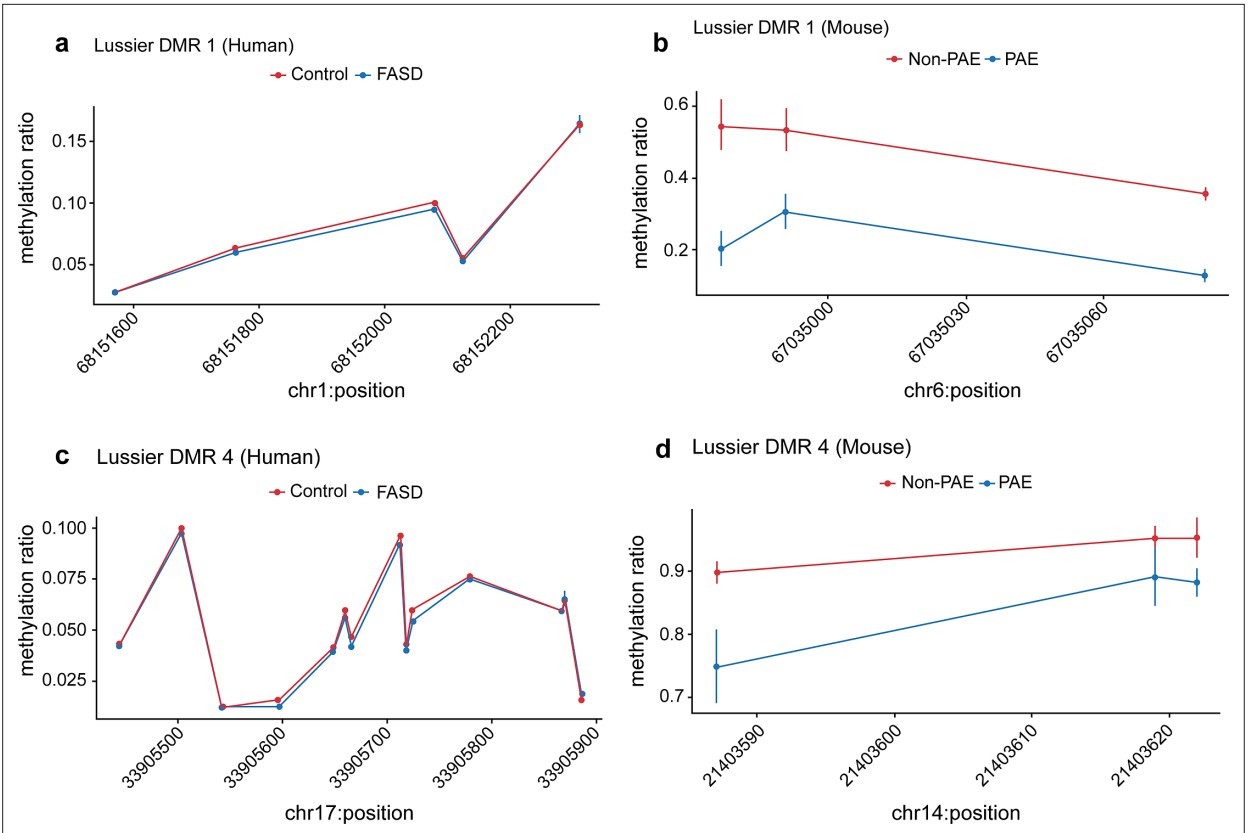

**Figure 9.** Seven prenatal alcohol exposure (PAE) differentially methylated regions (DMRs) identified in the murine model were successfully replicated in the *Lussier et al., 2018* human fetal alcohol spectrum disorder (FASD) cohort. Examples of two PAE DMRs that had significantly lower DNA methylation with a clinical diagnosis of FASD in the *Lussier et al., 2018* cohort (**a** and **c**), while their mouse liftover DMR also had significantly lower DNA methylation with PAE in the murine model experiment (**b** and **d**).

osmosis water and normal chow throughout pregnancy. Trajectories of liquid consumption and weight gain across pregnancy were assessed using a quadratic mixed effects model and the trajectory of chow consumption across pregnancy was assessed using a linear mixed effects model. To examine the effect of alcohol exposure on behavioural outcomes we used linear regression with alcohol group (binary) as the main predictor adjusted for diet and sex. For sequencing data, raw fastq files were mapped to the mm10 mouse reference genome with BSseeker 2 (version 2.1.8) (*Guo et al., 2013*) and CG-maptools (version number 0.1.2) (*Guo et al., 2018*) using a custom bioinformatics pipeline. CGmap output files were combined as a bsseq object in the R statistical environment (*R Development Core Team, 2021*). We filtered the sex chromosomal reads and then combined reads from mice in the same treatment group using the *collapseBSseq* function, to maximise coverage prior to differential DNA methylation analysis. CpG sites with an aggregated coverage below 10× in each tissue type were removed prior to modelling to ensure there was sufficient coverage in all assessed CpG sites. This retained 94.9% of CpG sites in the brain and 93.8% of CpG sites in the liver. DMRs were identified within each tissue using a Bayesian hierarchical model comparing average DNA methylation ratios in each CpG site between PAE and non-PAE mice using the Wald test with smoothing, implemented in the R package *DSS* (*Wu et al., 2015*). False discovery rate (FDR) control was achieved through shrinkage estimation methods. We declared DMRs as those with a local FDR p-value<0.05 based on the p-values of each individual CpG site in the DMR, and minimum mean effect size (delta) of 5%. Gene ontology analysis was performed on the brain and liver DMRs using the gene set enrichment analysis computational method (*Subramanian et al., 2005*) to determine if the DMRs were associated with any transcription start sites or biological processes. Brain and liver DMRs were tested for enrichment within ENCODE Project datasets (*Luo et al., 2020*) by an overlap permutation test with 100 permutations using the *regioneR* package. The ENCODE Project datasets that were assessed included ENCFF845WSI,

ENCFF764NTQ, ENCFF937JHP, ENCFF269TLO, ENCFF676TSV, and ENCFF290MLR. DMRs were then tested for enrichment within specific genic and CpG regions of the mouse genome, compared to a randomly generated set of regions in the mouse genome generated with *resampleRegions* in *regioneR*, with equivalent means and standard deviations. For candidate gene analysis, we compiled a set of key genes and genomic regions identified in previous mammalian PAE studies for site-specific testing based on our prior systematic review (*Bestry et al., 2022*), which identified 37 candidate genes (*Supplementary file 1e and f*). Murine brain and liver datasets were filtered to candidate gene regions and differential testing was then performed across the entire coding sequence, separately in the brain and liver of the mice on a normal diet using the *callDML* feature in DSS.

## Validation studies in human cohorts

We used existing human datasets to validate observations from our murine model (GSE109042), focusing on regions identified in our early moderate PAE model. Validation studies in human cohorts with existing genome-wide DNA methylation datasets and matching PAE data are described in the Supplementary Material. Briefly, Illumina Human Methylation array. iDAT files were pre-processed using the *minfi* package (*Aryee et al., 2014*) from the Bioconductor project (http://www.bioconductor.org) in the R statistical environment (http://cran.r-project.org/, version 4.2.2). Sample quality was assessed using control probes on the array. Between-array normalization was performed using the stratified quantile method to correct for Type 1 and Type 2 probe bias. Probes exhibiting a p-detection call rate of >0.01 in one or more samples were removed prior to analysis. Probes containing SNPs at the single base extension site, or at the CpG assay site were removed, as were probes measuring non-CpG loci (32,445 probes). Probes reported to have off-target effects in *McCartney et al., 2016*, were also removed. Mouse DMRs were converted into human equivalent regions using an mm10 to hg19 genome conversion with the liftover tool in the UCSC Genome Browser (*Kent et al., 2002*). A minimum 0.1 ratio of bases that must remap was specified as recommended for liftover between regions from different species and multiple output regions were allowed. Differential testing of candidate mouse DMRs was carried out using the R package *DMRcate* (*Peters et al., 2021*) for each dataset and DMRs were declared as minimum smoothed FDR<0.05. Cell heterogeneity in each sample including the composition of epithelial, fibroblast, and immune cells was estimated from DNA methylation reads using the R package *EpiDISH* (*Teschendorff et al., 2017*).

## Acknowledgements

We wish to acknowledge the assistance of Dr Jahnvi Pflueger who provided training on preparation of WGBS libraries. We wish to acknowledge the financial contribution of the Centre for Research Excellence in FASD who supported the murine experiments.

## Additional information

### Funding

| Funder | Grant reference number | Author |
|---|---|---|
| National Health and Medical Research Council | 114601 | James P Fitzpatrick<br>Elizabeth J Elliott<br>Martyn Symons |
| Department of Health, Government of Western Australia | WACRF2019/20 R8 | Alexander N Larcombe<br>James P Fitzpatrick<br>Elizabeth J Elliott<br>Jeffrey M Craig<br>Evelyne Muggli<br>Jane Halliday<br>Delyse Hutchinson<br>Sam Buckberry<br>Ryan Lister<br>Martyn Symons<br>David Martino |

| Funder | Grant reference number | Author |
|--------|------------------------|--------|

The funders had no role in study design, data collection and interpretation, or the decision to submit the work for publication.

## Author contributions

Mitchell Bestry, Data curation, Formal analysis, Investigation, Methodology, Writing - original draft, Project administration; Alexander N Larcombe, Conceptualization, Data curation, Formal analysis, Supervision, Funding acquisition, Methodology, Writing – review and editing; Nina Kresoje, Supervision, Methodology, Writing – review and editing; Emily K Chivers, Data curation, Methodology; Chloe Bakker, Data curation, Investigation, Methodology; James P Fitzpatrick, Elizabeth J Elliott, Jeffrey M Craig, Delyse Hutchinson, Funding acquisition, Writing – review and editing; Evelyne Muggli, Jane Halliday, Data curation, Funding acquisition, Writing – review and editing; Sam Buckberry, Ryan Lister, Supervision, Funding acquisition, Writing – review and editing; Martyn Symons, Conceptualization, Supervision, Funding acquisition, Writing – review and editing; David Martino, Conceptualization, Supervision, Project administration, Writing – review and editing

## Author ORCIDs

Mitchell Bestry ⓘ https://orcid.org/0000-0002-1962-7925
Jeffrey M Craig ⓘ https://orcid.org/0000-0003-3979-7849
Sam Buckberry ⓘ https://orcid.org/0000-0003-2388-6046
Ryan Lister ⓘ https://orcid.org/0000-0001-6637-7239
David Martino ⓘ https://orcid.org/0000-0001-6823-4696

## Ethics

This study received animal ethics approval from the Telethon Kids Institute Animal Ethics Committee (Approval Number: 344). All procedures were carried out in accordance with the approved protocol.

Reviewer #1 (Public Review): https://doi.org/10.7554/eLife.92135.3.sa1
Reviewer #2 (Public Review): https://doi.org/10.7554/eLife.92135.3.sa2
Author response https://doi.org/10.7554/eLife.92135.3.sa3

---

# Additional files

## Supplementary files

• Supplementary file 1. Statistical results from MWA studies. (a) Table of brain differentially methylated regions (DMRs) identified by DSS and annotated with annotatr. (b) Table of liver DMRs identified by *DSS* and annotated with *annotatr*. (c) Table of gene set enrichment analysis (GSEA) ontology results from genes associated with liver DMRs. (d) GSEA Ontology Gene/Gene Set Overlap Matrix for liver DMRs. (e) List of genes included in candidate gene analysis. (f) Table of regions assessed in candidate genes analysis. (g) Table of brain DMRs having differences to DNA methylation with prenatal alcohol exposure (PAE) being rescued by dietary supplementation. (h) Table of liver DMRs having differences to DNA methylation with PAE being rescued by dietary supplementation. (i) Table of FDR-significant brain DMLs from candidate gene regions in regular diet mice. (j) Table of FDR-significant liver DMLs from candidate gene regions in regular diet mice. (k) Sequencing statistics and bioinformatic quality control.

• MDAR checklist

## Data availability

The mouse WGBS data are deposited into the GEO database under accession number GSE273157.

The following dataset was generated:

| Author(s) | Year | Dataset title | Dataset URL | Database and Identifier |
|---|---|---|---|---|
| Martino DJ | 2024 | Early Moderate Prenatal Alcohol Exposure and Maternal Diet Impact Offspring DNA Methylation Across Species | https://www.ncbi.nlm.nih.gov/geo/query/acc.cgi?acc=GSE273157 | NCBI Gene Expression Omnibus, GSE273157 |

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
