## [Editor Report · eLife assessment]

This **important** study unveils the significant impact of prenatal alcohol exposure on epigenetic patterns, offering new insights into its adverse health outcomes through **solid** evidence from both mouse models and human data. The findings, which reveal how a high-methyl diet can mitigate these epigenetic alterations, present a promising prenatal care strategy. Despite its **solid** data overall, the study's small sample size and unaccounted confounders suggest the need for further research to confirm these findings and explore their practical implications.

---

## [Referee Report · Reviewer #1 (Public Review)]

Summary:

This manuscript examined the impact of prenatal alcohol exposure on genome-wide DNA methylation in the brain and liver, comparing ethanol-exposed mice to unexposed controls. They also investigated whether a high-methyl diet (HMD) could prevent the DNA methylation alterations caused by alcohol. Using bisulfite sequencing (n=4 per group), they identified 78 alcohol-associated differentially methylated regions (DMRs) in the brain and 759 DMRs in the liver, of which 85% and 84% were mitigated by the HMD group, respectively. The authors further validated 7 DMRs in humans using previously published data from a Canadian cohort of children with FASD.

Overall, the findings from this study provide new insight into the impact of prenatal alcohol exposure, while also showing evidence for methyl-rich diets as an intervention to prevent the effects of alcohol on the epigenome. Some methodological concerns and confounders limit the robustness of these results, and should be addressed in future studies to further strengthen the conclusions of this study and its applicability to broader settings.

Strengths:

- The use of whole genome bisulfite sequencing allowed for the interrogation of the entire DNA methylome and DMR analysis, rather than a subset of CpGs.

- The combination of data from animal models and humans allowed the authors to make stronger inferences regarding their findings

- The authors investigated a potential mechanism (high methyl diet) to buffer against the effects of prenatal alcohol exposure, which increases the relevance and applicability of this research.

Weaknesses:

- The sample size was small for the epigenetic analyses, which limits the robustness of the findings.

- The authors could not account for potential confounders in their analyses, including birthweight, alcohol levels, and sex. This is a particular problem for the high-methyl diet analyses, in which the alcohol-exposed mice consumed less alcohol than their non-diet counterparts.

---

## [Referee Report · Reviewer #2 (Public Review)]

Summary:

Bestry et al. investigated the effects of prenatal alcohol exposure (PAE) and high methyl donor diet (HMD) on offspring DNA methylation and behavioral outcomes using a mouse model that mimics common patterns of alcohol consumption in pregnancy in humans. The researchers employed whole-genome bisulfite sequencing (WGBS) for unbiased assessment of the epigenome in the newborn brain and liver, two organs affected by ethanol, to explore tissue-specific effects and to determine any "tissue-agnostic" effects that may have arisen prior to the germ-layer commitment during early gastrulation. The authors found that PAE induces measurable changes in offspring DNA methylation. DNA methylation changes induced by PAE coincide with non-coding regions, including enhancers and promoters, with the potential to regulate gene expression. Though the majority of the alcohol-sensitive differentially methylated regions (DMRs) were not conserved in humans, the ones that were conserved were associated with clinically relevant traits such as facial morphology, educational attainment, intelligence, autism, and schizophrenia Finally, the study provides evidence that maternal dietary support with methyl donors alleviates the effects of PAE on DNA methylation, suggesting a potential prenatal care option.

Strengths:

The strengths of the study include the use of a mouse model where confounding factors such as genetic background and diet can be well controlled. The study performed whole-genome bisulfite sequencing, which allows a comprehensive analysis of the effects of PAE on DNA methylation.

Weaknesses:

Transcriptome analysis to test if the identified DMRs indeed affect gene expression would help determine the potential function of the identified methylation changes.

---

## [Author Response]

The following is the authors’ response to the original reviews.

Response to reviewers

We wish to thank the reviewers for the time taken to appraise the manuscript and the helpful feedback to improve it. We have taken onboard the suggested feedback and incorporated it into the revision. The findings of the revised manuscript are unchanged. Below is a point-by-point response to specific comments.

**Public reviews**

**Reviewer 1**

Thank you to reviewer 1 for the thorough and insightful review of our manuscript. We are pleased that the strengths of our research, particularly the use of whole-genome bisulfite sequencing, the combination of animal and human data, and the investigation of a potential dietary intervention were recognized. We are confident that these aspects contribute significantly to the value and originality of our work.

We acknowledge the concerns regarding the statistical rigor of the study, particularly the sample size and data analysis methods. We would like to address these points in more detail:

Sample size: While we agree that a larger sample size would be ideal, the chosen sample size (n=4 per group) is consistent with other murine whole-genome bisulfite sequencing experiments in the field. We have carefully considered the cost-benefit trade-off in selecting this approach. In the revision we discuss the potential limitations of this sample size.

Data analysis: We acknowledge the inconsistencies in the study reporting and have committed to improving the clarity in the revision. We carefully reviewed the concerns regarding the use of causal language and the interpretation of differences in our results. In some cases, the use of causal language is justified by the intervention study design. We also believe other explanations like stochastic variation affecting the same genomic regions in different tissues, are exceedingly unlikely from a statistical viewpoint. In the revision we have adopted a balanced approach to the language.

Confounders: We acknowledge the importance of accounting for potential confounders such as birthweight, alcohol exposure and sex. The pups selected for genome analysis were matched for sex and on litter size as a proxy for in utero alcohol exposure. This careful selection of mice for genome analysis was intentionally guided to mitigate potential confounding.

Statistical rigour: We acknowledge the importance of multiple testing correction in the genome-wide analysis. We used the DSS method of Feng et al (PMID: 2456180) which employs a two-step procedure for assessing significance of a region. Instead of a single p-value for the whole DMR, DSS uses the area statistic to rank candidate regions and control the false discovery rate through shrinkage estimation methods. This approach reduces the risk of reporting false positives due to multiple testing across numerous CpG sites. It is similar in respects to employing local FDR correction at 0.05 level, with an additional minimum effect size threshold applied, and particularly suited to experiments where the number of replicates is low. In the revision we have committed to improving the clarity of the reporting of statistical methods.

**Reviewer 2**

Thank you to reviewer 2 for the comprehensive and valuable feedback on our manuscript. We take your concerns about the generalizability of our findings and the interpretation of certain results seriously. We would like to address your specific criticisms in detail:

Generalizability and Human Data: We agree that the generalizability of mouse models to human conditions has limitations. However, our study focused on understanding the early molecular alterations caused by moderate PAE, which can be more effectively modelled in a controlled environment like mice. To clarify this, we have strengthened the manuscript by emphasizing the focus on moderate PAE in the title and throughout the paper.

Transcriptome Analysis: We recognize the importance of investigating the functional consequences of PAE-induced DMRs and agree that transcriptome analysis would be highly valuable. We are currently planning to conduct future transcriptomic studies to understand the link between DMRs and gene expression.

Species-Specificity and DMR Enrichment: We acknowledge the likelihood of species-specific PAE effects. Our finding of enrichment of DMRs in non-coding regions was consistent with observations from the Lussier study of FASD. We agree there is further work to do and now highlight this in the discussion.

Tissue Sample Locations: Due to technical restrictions of processing newborn mouse tissue, we are unable to enhance the manuscript with specific tissue regions sampled.

Interpretation of Shared Genomic Regions: We appreciate your point about the alternative explanation for the shared genomic regions between brain and liver. Our interpretation is that regions identified in the alcohol group only affected equally in both tissues are likely established stochastically (as a result of the exposure) in the early embryo and then maintained in the germ layers. We have revised to suggest this is the most likely explanation and we acknowledge a more detailed examination in more tissues would be warranted for proof.

Additional Feedback

**Reviewer 1**
IntroductionLine 65 - alcohol consumption is not always preventable and these statements further increase the stigma associated with FASD. A better way to say this would be "a leading cause of neurodevelopmental impairments".

We have implemented this suggestion in revised manuscript.

The studies cited in lines 87-89 are somewhat outdated, as several more recent studies with better sample sizes have been published in recent years. I would recommend citing more recent publications in addition to these studies. Similarly, the authors should also cite Portales-Casamar et al., 2016 (Epigenetic & Chromatin) for the validation in humans, as it was the original study for those data.

We have added a citation for the study mentioned by Portales-Casamar et al. (2016) in the revised manuscript.

Lines 95-95 - the authors should elaborate further on the "encouraging results" from choline supplementation studies, as these details may help interpret the findings from their own study.

In the revised manuscript, we replaced “encouraging results” with “results suggesting a high methyl donor diet (HMD) could at least partially mitigate the adverse effects of PAE on various behavioural outcomes”.

Minor point: DNA methylation is preferable to "methylation" alone when not referring to specific CpGs or sites, as methylation can also refer to protein or RNA methylation.

“Methylation” has been replaced with “DNA methylation” in revised manuscript

ResultsLine 118 - HMD should be defined here.

HMD defined in revised manuscript

The figures in the main manuscript and supplemental materials are not in the same order as they are presented in the text.

We apologise for this and thank the reviwer for their attendtion to detail. In the revision we have corrected the order of figures to match the text.

It is concerning that the H20-HMD group had lower baseline weights, which could impact the findings from these analyses. Please discuss how these differences were accounted for in the study design and analyses.

We appreciate the reviewer's concern about the lower baseline weight in the H20-HMD group. We agree that this difference could potentially affect our findings. However, we want to emphasize that total weight gain during pregnancy was statistically similar across all groups by linear mixed effect model. Additionally, all dams were within the healthy weight range for their strain. While we cannot completely rule out any potential influence of baseline weight, we believe the similarity in weight gain and the healthy range of all dams suggest that the in-utero experience of pups regarding weight-related factors was likely comparable across groups.

I have some concerns regarding the cutoffs used to identify the DMRs, particularly given the small N and number of tests. The authors should report the number of DMRs that meet a multiple testing threshold; if none, they should use a more stringent threshold than p<0.05, as one would expect 950,000 CpGs to meet that threshold by chance (19,000,000 CpGs x 0.05). The authors should also report the number of DMRs tested, as this will be a more appropriate benchmark for their analyses than the number of CpGs (they should also report the specific number here).

We appreciate the reviewer's concerns regarding the DMR cut-offs. We agree that clarifying the methods and justifying our choices is crucial. Our implementation of the DSS method for defining DMRs employs a local FDR p<0.05 cut-off, with additional delta beta threshold of 5%. We have clarified this in the methods section of the revised manuscript . We want to emphasize that the local FDR approach effectively mitigates the concern of chance findings by adjusting for multiple comparisons across the genome. Line 414-420 in the revised methods contains the following amended text

“Differentially methylated regions (DMRs) were identified within each tissue using a Bayesian hierarchical model comparing average DNA methylation ratios in each CpG site between PAE and non-PAE mice using the Wald test with smoothing, implemented in the R package DSS (46). False-discovery rate control was achieved through shrinkage estimation methods. We declared DMRs as those with a local FDR P-value < 0.05 based on the p-values of each individual CpG site in the DMR, and minimum mean effect size (delta) of 5%”

I also have concerns about the delta cutoff for their DMRs. First, it is not clear if this cutoff is set for a single CpG or across the DMR (even then, it is not clear if this is a mean, median, max, min, etc.) Second, since the authors analyzed CpGs with 10X coverage, they can only reliably detect a delta of 0.1 (1/10 reads).

Thank you for raising this important point. In the revision we have clarified the effect size cutoff reflects the mean effect across CpGs within the DMR as follows (line 418)

“We declared DMRs as those with a local FDR P-value < 0.05 based on the p-values of each individual CpG site in the DMR, and minimum mean effect size (delta) of 5%”

We chose the mean as it provides a comprehensive representation of the overall methylation change within the region, while ensuring all individual CpGs used in the analysis had at least 10x coverage. It is not true that we can only detect a delta of 1/10 reads, the mean effect is the relative difference in means between groups and is not dependent on the underlying sequencing depth.

Prenatal alcohol exposure is known to impact cell type proportions in the brain, which could lead to differences in DNAm patterns. The authors should address this possibility in the discussion, as well as examine their list of DMRs to determine if they are associated with specific brain cell types. The possibility of cell type differences in the liver should also be discussed.

We agree with the reviewer that PAE-induced alterations in cell type proportions can influence DNA methylation patterns. While isolating specific cell types in our current study's brain and liver samples was not achievable due to tissue limitations, we acknowledge this as a limitation and recognize the need for further investigations incorporating single-cell or cell type-specific approaches in the discussion.

It is interesting, but maybe not surprising, that more DMRs were identified in the liver compared to the brain. This finding would warrant some additional interpretation in the discussion.

We appreciate and agree that this finding indeed warrants further interpretation. We have added the following sentence into the discussion section of the revised manuscript that provides some potential factors behind this observation.

Lines 263 “Indeed, most of the observed effects were tissue-specific, with more perturbations to the epigenome observable in liver tissue, which may reflect the liver’s specific role in metabolic detoxification of alcohol. Alternatively, cell type composition differences between brain and liver might explain differential sensitivity to alcohols effects”.

Lines 148-149 - I disagree about the enrichment of decreased DNAm in brain DMRs, as 52.6% is essentially random chance. The authors should also include a statistical test here, such as a chi-squared test, to support this statement.

We agree that a revised interpretation is warranted. The updated manuscript has been amended as follows: “Lower DNA methylation with early moderate PAE in NC mice was more frequently observed in liver DMRs (93.5% of liver DMRs), while brain DMRs were almost equally divided between lower and higher DNA methylation with early moderate PAE (52.6% of brain DMRs had lower DNA methylation with early moderate PAE).”

Similarly, I would recommend the authors use increased/decreased DNAm, rather than hypermethylated/hypomethylation, as the latter terms are better suited to DNAm values near 100% or 0%.

The use of hyper/hypo methylation is still considered common and well understood even for moderate changes. We agree the use of increased/decreased is more inclusive for a broader audience, so we have amended all references accordingly in the main text.

Lines 153-155 - please report the statistics to support these enrichment results. A permutation test would be well suited to this analysis.

The reporting of statistics related to the enrichment test has now been amended to read “Overlap permutation tests showed liver DMRs were enriched in inter-CpG regions and non-coding intergenic regions (p < 0.05), while being depleted in all CpG regions and genic regions except 1to5kb, 3UTR and 5UTR regions, where there was no significant difference (Figure 2f).”

Line 156 - "overwhelming enrichment" is a very strong statement considering the numbers themselves.

Omitted “overwhelming” in revised manuscript. Revised manuscript states: “Using open chromatin assay and histone modification datasets from the ENCODE project, we found enrichment (p < 0.05) of DMRs in open chromatin regions (ATAC-seq), enhancer regions (H3K4me1), and active gene promoter regions (H3K27ac), in mouse fetal forebrain tissue and fetal liver (Table 2).”

Lines 165-167 - Please describe the analyses and metrics used to determine if the DNAm differences were mitigated in the HMD groups. As it stands, it is not clear if they are simply not significant, or if the delta was decreased. In terms of a figure, a scatter plot of the deltas for these DMRs would be better suited to visualizing these changes.

To determine whether DMRs were mitigated we simply applied the same statistical testing procedure on the subset of PAE DMRs in the group of mice exposed to the HM diet. The sample size is the same, and the burden on multiple testing is reduced as we did not test the entire genome. We believe our interpretation stands although we have urged caution in the discussion as follows (line 319)

“Another key finding from this study was that HMD mitigated some of the effects of PAE on DNA methylation. Although a plausible alternative explanation is that some of the PAE regions were not reproduced in the set of mice given the folate diet, our data are consistent with preclinical studies of choline supplementation in rodent models (34, 35) (36). Moreover, a subset of PAE regions were statistically replicated in subjects with FASD, suggestive or robust associations. Although our findings should be interpreted with caution, they collectively support the notion that alcohol induced perturbation of epigenetic regulation may occur, at least in part, through disruption of the one-carbon metabolism.”

Given the lenient threshold to identify DMRs, it is possible that PAE-associated DMRs are simply false positives and do not "replicate" in a different subset of animals. One way to check this would be to determine whether there are any differences between mitigated/unmitigated DMRs and the strength of their initial associations. Should the mitigated DMRs skew towards higher p-values and lower deltas, one might consider that these findings could be false positives.

We appreciate the reviewer's concern about potential false positives due to the chosen DMR identification threshold. We reiterate the DMR calling thresholds were adjusted for local FDR; however, we acknowledge the need for further validation. We haven't observed this trend of mitigated DMRs having higher p-values and lower deltas, but we have replicated some PAE DMRs in independent human datasets and found support for their biological plausibility in the context of PAE.

Related to the HMD analyses, I am concerned that the EtOH-HMD group consumed less alcohol, which could manifest in the PAE-induced DMRs disappearing, unrelated to the HMD exposure. The authors should comment on whether the pups were matched for ethanol exposure and include sensitivity analyses that include ethanol level as a covariate to confirm that their results are not simply due to decreased alcohol exposure.

We appreciate the reviewer's concern regarding the lower alcohol consumption by Dams in the EtOH-HMD group and its potential impact on DMRs. We agree that consistent in utero exposure is crucial for reliable results. Our pup selection for genomic analysis involved matching litter size as a proxy for in utero exposure, so even through the average alcohol consumption was lower for the EtOH-HMD group, we matched pups across treatment groups based on litter size as a proxy for alcohol intake levels, excluding pups with significantly different exposure levels. We agree more robust methods including direct measurement of blood alcohol content would improve the study. We have now incorporated this into the discussion of the revised manuscript on lines 351: “Additionally, we employed an ad-libitum alcohol exposure model rather than direct dosing of dams. Although the trajectories of alcohol consumption were not statistically different between groups, this introduces more variability into alcohol exposure patterns, and might might impact offspring methylation data”

Lines 172 - please be more specific about the neurocognitive domains tested.

In the revision we have included more detail about the neurocognitive domains tested (originally mentioned in the results) in the methods as follows:

“These tests included the open field test (locomotor activity, anxiety) (38), object recognition test (locomotor activity, spatial recognition) (39), object in place test (locomotor activity, spatial recognition) (40), elevated plus maze test (locomotor activity, anxiety) (41), and two trials of the rotarod test (motor coordination, balance) (42)”

Line 191 - please report the tissue type used in the human study, as well as the method used to estimate cell type proportions.

We stated in the results section that buccal swabs were used in both human cohorts.

We added to the revised manuscript that cell type proportions were estimated using the EpiDISH R package.

Related to validation, it is unclear whether the human-identified DMRs were also validated in mice, or if the authors are showing their own DMRs. Please also discuss why DMRs might not have been replicated in AQUA.

We used human data sets to validate observations from our murine model, focusing on regions identified in our early moderate PAE model. This is now explicitly state on line 209 of the revision:

“We undertook validation studies by examining PAE sensitive regions identified in our murine model using existing DNA methylation data from human cohorts to address the generalizability of our findings.”

“In the section entitled ‘Candidate Gene Analysis..’ we used our murine data sets to reproduce previously published associations that included regions identified in both animal and human studies. We posit the lack of replication of our early moderate PAE regions in AQUA is explained in part by species-specific differences and considering the striking differences in effect size seen in regions that did replicate in FASD subjects, the exposure may need to be of sufficient magnitude and duration for the effects seen in brain and liver to survive reprogramming in the blood. The AQUA cohort is largely enriched for low to moderate patterns of alcohol consumption.

Line 197 - please provide a citation for the ethanol-sensitive regions. There are also several existing DNAm analyses in brain tissues from animal models that should be included as part of these analyses, as several have shown brain-region and sex-specific DMRs related to prenatal alcohol exposure. These contrasts might help the authors further delineate the effects of prenatal alcohol in their model and expand on current literature to explain the deficits caused by alcohol exposure.

Our candidate gene/region selection was informed by a systematic review of previously published human and animal studies reporting associations between in utero exposure to PAE and offspring DNA methylation. We synthesized evidence across several models, tissues and methylation platforms to arrive at a core set of reproducible associations. Line 481 of the methods now includes a citation to our systematic review which details our selection criteria.

DiscussionLine 211 - This is a strong statement for one hypothesis. It is also possible that different cell types have similar responses to prenatal alcohol exposure. In this scenario, perturbations need not arise before germ layer separation. The authors should soften this causal statement.

We appreciate this point although given the genome size relative to the size of the DMRs we have detected, the chance that different cell types would respond similarly in exactly the same regions seems exceedingly rare. We posit a more likely explanation is early perturbations in the embryo are established stochastically as a result of the exposure (supported by the interventional design) and maintained in the differentiating tissues. We agree further work is needed to prove this, specifically in a wider set of tissues from multiple germ layers so we have amended the discussion as follows:

“These perturbations may have been established stochastically because of alcohol exposure in the early embryo and maintained in the differentiating tissue. Further analysis in different germ layer tissues is required to formally establish this.”

Lines 222-224 - I completely agree with this statement. However, the authors had the opportunity to examine dosage effects in their model as they measured alcohol-levels from the dams. At the very least, I would recommend sensitivity analyses in their DMRs to assess whether alcohol level/dosage influences their results.

Although a great suggestion to improve the manuscript, we did not have opportunity to examine dosages by design as we selected mice for genome analysis with matched exposure patterns. It would be fascinating to conduct a sensitivity analysis.

Methods:Please include the lysis protocol.

Thank you for picking up this error in our reporting. We have now included the following details in the methods which improve the reproducibility of this study:“Ten milligrams of tissue were collected from each liver and brain and lysed in Chemagic RNA Tissue10 Kit special H96 extraction buffer”.

Please include the total reads for each sample and details of the QC pipeline, including filtering flags, quality metrics, and genome build.

Thank you for suggesting improvements to our reporting which improve the reproducibility of this study. We have included a new supplementary tableTab of sequencing statistics and details of the quality metrics. Please note the genome build is explicitly stated in the methods already.

Please make your code publicly available to ensure that these analyses can be replicated.

Thank you for this suggestion. A data availability statement has now been included in the revision and code will be made available upon request

Why were Y chromosome reads included in the dataset?

Y chromosomal reads were not included in the DMR analysis. Amended “We filtered the X chromosomal reads” to “We filtered the sex chromosomal reads” in revised manuscript.

Please provide the number of total CpGs available for analysis.

Added sentence into results section of revised manuscript: “A total of 21,842,961 CpG sites were initially available for analysis.” We also clarified that the ~19,000,000 CpGs were analysed following coverage filtering.

Please provide the parameters for the DMR analysis and report how the p-values and deltas were calculated.

We have addressed this in previous comments

The supplemental materials for the human data are missing.

Thank you for picking up this oversight. The revision now includes an additional data supplement which details the analysis of the human data sets for interested readers.

Tables and figuresTable 1. It is not clear how the DMRs for this table were selected. The exact p-values and FDR should also be reported in this table. The number of CpGs in these DMRS should also be reported.

Table 1 includes select DMRs that were consistently detected in both brain and liver tissue. These are particularly of interest as they represent regions highly sensitive to alcohol exposure. We agree that exact reporting of p-values would be ideal. Instead of a single p-value for the whole DMR, DSS uses the area statistic to rank candidate regions and control the false discovery rate (FDR) through shrinkage estimation methods. In the revision we have now included region size and number of CpGs in table 1.

Table 3. Please include p-values for the DMR analyses.

As above we report the area-statistic which is an equivalent measure to assess evidence for differential methylation.

Figure 2 (Figure 4 in revised manuscript). Please report the N for these analyses. It also seems that the pairwise t-tests were only compared to the H20-NC, which does not provide much insight into the PAE group. The relevance of the sexP analysis to the present manuscript is also unclear.

Figure 2 is now Figure 4 in the revision and the sample size has been included in figure legend. We compared all groups to the control group (H20-NC) as we aimed to determine any differences in intervention groups from the control.

We apologies for lack of clarity around the ‘sex P’ terminology. This refers to the p-value for the main effect of sex on the behavioural outcome. We agree it lacks relevance since the regression models were adjusted for sex. In the revision we have updated the methods as follows (line426) and removed references to sex P

“To examine the effect of alcohol exposure on behavioural outcomes we used linear regression with alcohol group (binary) as the main predictor adjusted for diet and sex.”

Figure 3ef (Figure 2ef in revised manuscript). It is unclear how the regions random regions were generated. A permutation test would be relevant to determine whether there are any actual enrichment differences.

As stated in methods section: “DMRs were then tested for enrichment within specific genic and CpG regions of the mouse genome, compared to a randomly generated set of regions in the mouse genome generated with resampleRegions in regioneR, with equivalent means and standard deviations.”

Figure 5. Please include the gene names for these DMRs, as well as their genomic locations. It would also be relevant to annotate these plots with the max, min, and mean delta between groups.

Thank you, we considered this however the DMRs are not in genes so we cannot apply a gene label. The locations are reported on the x-axis and the statistics are shown in Table 3.

Figure S1b and S2c- It is quite worrisome that the PAE-HMD group drank less throughout pregnancy than their PAE counterparts. Please discuss how this was addressed in the analyses.

We appreciate the reviewer's concern regarding the lower alcohol consumption in the PAE-HMD group and its potential impact on DMRs. We agree that consistent in-utero exposure is crucial for reliable results. Although the total amount of liquid consumed over pregnancy was lower in this group, they started with a lower baseline and the trajectory was not statistically different compared to other groups.

We have now incorporated this into the discussion section of the revised manuscript on lines 336: “Additionally, we employed an ad-libitum alcohol exposure model rather than direct dosing of dams. Although the trajectories of alcohol consumption were not statistically different between groups, this introduces more variability into alcohol exposure patterns, and might might impact offspring methylation data.”

Figure S1cd. See my comments about Figure 2.

Suggested changes have been incorporated.

Figure S2d. it is not clear to what the statistics presented in this panel refer. Please clarify and discuss the implications of dietary intake differences on your findings.

Added sentence to caption in revised manuscript: “Statistical analysis involved linear mixed-effects regression comparing trajectories of treatment groups to H2O-NC baseline control group.”

Figure S3. See my comments about Figure 2.

Suggested changes have been incorporated

Figure S4. I am confused by the color legend, as it seems both colors are PAE. I also do not see how any regions show increased or decreased DNAm in PAE based on this plot (also no statistics are presented to support these conclusions).

The plot is intended to show there are no gross changes in methylation when averaged across all CpGs within different regulatory genomic contexts. Statistics are not included as it is intuitive from the plot that the means are the same. We have updated the figure legend which now reads

“Figure S4. No evidence for global disruption of methylation by PAE. The figure shows methylation levels averaged across CpGs in different regulatory genomic contexts. Neither brain tissue (A & B), nor liver tissue (C & D) were grossly affected by PAE exposure (blue bars). Bars represent means and standard deviation.”